# Synaptic and peptidergic connectome of a neurosecretory center in the annelid brain

Elizabeth A Williams[1], Csaba Verasztó[1], Sanja Jasek[1], Markus Conzelmann[1], Réza Shahidi[1,2], Philipp Bauknecht[1], Olivier Mirabeau[3], Gáspár Jékely[1,2]*

[1]Max Planck Institute for Developmental Biology, Tübingen, Germany; [2]Living Systems Institute, University of Exeter, Exeter, United Kingdom; [3]Genetics and Biology of Cancers Unit, Institut Curie, INSERM U830, Paris Sciences et Lettres Research University, Paris, France

**Abstract** Neurosecretory centers in animal brains use peptidergic signaling to influence physiology and behavior. Understanding neurosecretory center function requires mapping cell types, synapses, and peptidergic networks. Here we use transmission electron microscopy and gene expression mapping to analyze the synaptic and peptidergic connectome of an entire neurosecretory center. We reconstructed 78 neurosecretory neurons and mapped their synaptic connectivity in the brain of larval *Platynereis dumerilii*, a marine annelid. These neurons form an anterior neurosecretory center expressing many neuropeptides, including hypothalamic peptide orthologs and their receptors. Analysis of peptide-receptor pairs in spatially mapped single-cell transcriptome data revealed sparsely connected networks linking specific neuronal subsets. We experimentally analyzed one peptide-receptor pair and found that a neuropeptide can couple neurosecretory and synaptic brain signaling. Our study uncovered extensive networks of peptidergic signaling within a neurosecretory center and its connection to the synaptic brain.

DOI: https://doi.org/10.7554/eLife.26349.001

**\*For correspondence:**
g.jekely@exeter.ac.uk

**Competing interests:** The authors declare that no competing interests exist.

## Introduction

Nervous system signaling occurs either at synapses or via secreted diffusible chemicals that signal to target cells expressing specific receptors. Synapse-level connectomics using electron microscopy allows mapping synaptic networks, but fails to reveal non-synaptic signaling. In addition to acting in a neuroendocrine fashion, non-synaptic volume transmission by neuropeptides and monoamines can have neuromodulatory effects on synaptic signaling (*Bargmann, 2012*; *Marder, 2012*; *Nusbaum et al., 2017*; *Jékely et al., 2017*). Overlaying synaptic and peptidergic maps is challenging and requires knowledge of the expression of the modulators and their specific receptors as well as synaptic connections. Such mapping has only been achieved for relatively simple circuits, such as the stomatogastric nervous system of crustaceans where synaptic connections are known and the effect of neuropeptides on the activity of single neurons can be analyzed experimentally (*Stein et al., 2007*; *Thirumalai and Marder, 2002*; *Blitz et al., 1999*). Likewise, connectome reconstructions combined with cellular-resolution neuropeptide and receptor mapping allow the dissection of peptidergic signaling in *Drosophila* (*Schlegel et al., 2016*; *Diao et al., 2017*). In *Caenorhabditis elegans*, the spatial mapping of monoamines and neuropeptides and their G protein-coupled receptors (GPCRs) revealed interconnected networks of synaptic and non-synaptic signaling (*Bentley et al., 2016*).

Neurosecretory centers, such as the vertebrate hypothalamus and pituitary, are found in the anterior brain of many animals and show exceptionally high levels of neuropeptide expression (*Herget and Ryu, 2015*; *Siegmund and Korge, 2001*; *Campbell et al., 2017*) suggesting extensive non-synaptic signaling. These centers coordinate many processes in physiology, behavior, and development, including growth, feeding, and reproduction (*Sakurai et al., 1998*; *Bluet-Pajot et al.,*

*2001*; *Sternson et al., 2013*). The combined analysis of synaptic and peptidergic networks is particularly challenging using single marker approaches for neurosecretory centers that express dozens of neuropeptides. To understand their function, a global mapping of peptidergic networks within these brain regions and how they connect to the rest of the nervous system is required.

Here, we analyze synaptic and peptidergic signaling in the anterior neurosecretory center in larval *Platynereis dumerilii*, a marine annelid. Annelid and other marine larvae have an anterior sensory center, the apical organ, involved in the detection of various environmental cues (*Hadfield et al., 2000*; *Conzelmann et al., 2013a*; *Page, 2002*; *Chia and Koss, 1984*). The apical organ is neurosecretory and expresses diverse neuropeptides that are thought to regulate various aspects of larval behavior and physiology, including the induction of larval settlement and metamorphosis (*Mayorova et al., 2016*; *Thorndyke et al., 1992*; *Tessmar-Raible et al., 2007*; *Conzelmann et al., 2011*; *Marlow et al., 2014*). Apical organs have a conserved molecular fingerprint across marine larvae, suggesting that they represent a conserved sensory-neuroendocrine structure (*Marlow et al., 2014*). The apical organ area or apical nervous system (ANS) in *Platynereis* larvae shows a distinct molecular fingerprint with similarities to other neuroendocrine centers, including the anterior medial neurosecretory center of arthropods and the vertebrate hypothalamus, suggesting a common ancestry (*Tessmar-Raible et al., 2007*; *Steinmetz et al., 2010*; *Conzelmann et al., 2013b*; *Hunnekuhl and Akam, 2014*). Molecular and developmental similarities in various protostomes and deuterostomes further suggest a more widespread conservation of neuroendocrine centers (*Hartenstein, 2006*; *Wirmer et al., 2012*; *Tessmar-Raible, 2007*; Hunnekuhl and Akam 2014). The study of marine invertebrate larval apical organs could thus inform about the evolution of neuroendocrine cell types and signaling mechanisms in metazoans.

*Platynereis* larvae represent a powerful system to analyze gene expression and synaptic connectivity in a whole-body context, allowing linking distinct neuropeptides and other molecules to single neurons (*Asadulina et al., 2012*; *Williams and Jékely, 2016*; *Shahidi et al., 2015*; *Achim et al., 2015*; *Pettit et al., 2014*). To understand how synaptic and peptidergic signaling is integrated in the *Platynereis* ANS, we combine serial section electron microscopy with the cellular analysis of neuropeptide signaling. This combined analysis revealed extensive non-synaptic peptidergic signaling networks within the ANS distinguishing this area from the rest of the nervous system. Through connectomics and functional studies we also reveal how this endocrine region can interact with the synaptic nervous system by peptidergic modulation of the ciliomotor circuitry.

## Results

### Ultrastructural reconstruction of the anterior neurosecretory center

To comprehensively map a neurosecretory center with ultrastructural detail, we focused on the larvae of the marine annelid *Platynereis*. Due to their small size, the larvae are amenable to whole-body connectomic analysis (*Randel et al., 2015*; *Shahidi et al., 2015*). We used a previously reported full-body serial-section transmission electron microscopy (ssTEM) dataset of a 3-day-old larva (*Randel et al., 2015*) and reconstructed its entire anterior neurosecretory nervous system (*Figure 1A–D*). *Platynereis* and other annelids have an anterior neurosecretory plexus containing the projections of peptidergic sensory-neurosecretory neurons (*Tessmar-Raible et al., 2007*; *Aros et al., 1977*). The neurosecretory plexus forms an anatomically and ultrastructurally distinct area that can be clearly distinguished from other neuropils, including the adjacent optic and nuchal organ neuropils (*Randel et al., 2014*; *Shahidi et al., 2015*)(*Figure 1D*). Neurites in this area have a high number of dense-core vesicles and very few synapses (*Figure 1E,F*). Classic neurotransmitter synapses can be identified by large clusters of clear synaptic vesicles in the axons, while peptidergic synapses appear as smaller clusters of dense core vesicles (*Randel et al., 2014*; *Shahidi et al., 2015*). We reconstructed all neurons that project to this region (*Figure 1B* and *Figure 1—figure supplement 1*, *Video 1*) and identified 70 sensory neurons and eight projection interneurons, the latter project in and out of the neurosecretory plexus. Most of the sensory neurons are bilaterally symmetric pairs with distinct axonal projection patterns, except for a few asymmetric neurons (*Figure 1—figure supplement 1*). The sensory neurons have diverse apical sensory specializations. Based on these morphological criteria we could distinguish at least 20 different sensory cell types with likely different sensory functions. For example, there are four ciliary photoreceptor cells (cPRC)

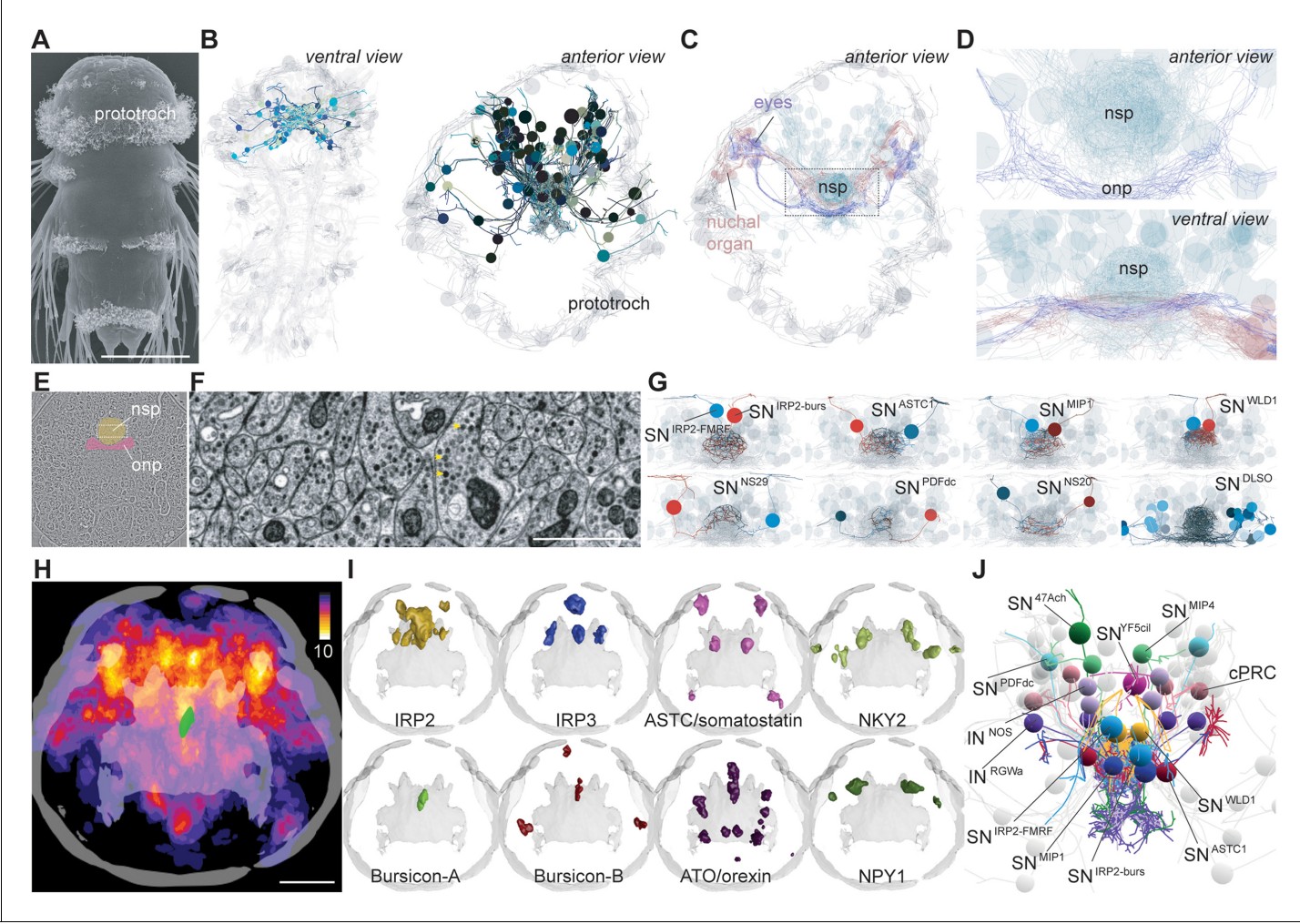

**Figure 1.** EM reconstruction and mapping of proneuropeptide expression in the apical nervous system of *Platynereis* larvae. (**A**) Scanning electron micrograph of a 3-day-old *Platynereis* larva. (**B**) Reconstructed apical nervous system neurons (ANS) (shades of blue) in a full-body serial section transmission electron microscopy (ssTEM) dataset of a 3-day-old larva, shown against a framework of reconstructed ciliated cells and axonal scaffold (light grey). Axons and dendrites appear as lines and cell body positions are represented by spheres. (**C**) ANS neurons project to an apical neurosecretory plexus in the center of the head, which forms a small sphere dorsal and apical to the optic neuropil and nuchal organ neuropil. (**D**) Close-up view of neurosecretory plexus indicated by box in C. (**E**) TEM image of a section in the larval head. The neurosecretory plexus and the optic neuropil are highlighted. (**F**) Close-up view of the neurosecretory plexus indicated by boxed area in E. Yellow arrowheads point at dense core vesicles. (**G**) Examples of bilaterally symmetric pairs of sensory neurons that innervate different regions of the neurosecretory plexus in the reconstructed ANS (grey), ventral view. (**H**) Heat map of the expression of 58 proneuropeptides in the 3-day-old *Platynereis* larva, apical view, projected on a reference scaffold of cilia and axonal scaffold (grey). Color scheme indicates number of different proneuropeptides expressed in a cell. An apical bursicon-A-expressing neuron of the ANS (SN$^{IRP2\text{-}burs}$) is overlaid for spatial reference (green). (**I**) Examples of average gene expression patterns of individual proneuropeptides from the 3 day proneuropeptide expression atlas. (**J**) ssTEM reconstruction of the 3-day-old larval ANS showing neurons that could be assigned specific proneuropeptide expression based on position and sensory cilia morphology. Abbreviations: nsp, apical neurosecretory plexus; onp, optic neuropil. Scale bars: A, 70 μm; H, 30 μm; F, 1 μm.

DOI: https://doi.org/10.7554/eLife.26349.002

The following source data and figure supplements are available for figure 1:

**Source data 1.** Reconstructed skeletons of *Platynereis* neurons in the 3-day-old larva ssTEM data stack have been submitted to the NeuroMorpho database http://neuromorpho.org/dableFiles/Jekely/Supplementary/Williams_Archive.zip.

DOI: https://doi.org/10.7554/eLife.26349.012

**Figure supplement 1.** TEM reconstruction of ANS neurons.

DOI: https://doi.org/10.7554/eLife.26349.003

**Figure supplement 2.** Average gene expression patterns of individual proneuropeptides from a whole-body gene expression atlas from 2-day-old larvae, anterior view.

DOI: https://doi.org/10.7554/eLife.26349.004

*Figure 1 continued on next page*

*Figure 1 continued*

**Figure supplement 2—source data 1.** Blender gene expression atlas for 2-day-old *Platynereis* larvae, including gene expression patterns of 53 neuropeptide precursor genes.

DOI: https://doi.org/10.7554/eLife.26349.005

**Figure supplement 3.** Average gene expression patterns of individual proneuropeptides from a whole-body gene expression atlas from 3-day-old larvae, anterior view.

DOI: https://doi.org/10.7554/eLife.26349.006

**Figure supplement 3—source data 1.** Blender gene expression atlas for 3-day-old *Platynereis* larvae, including gene expression patterns of 58 neuropeptide precursor genes.

DOI: https://doi.org/10.7554/eLife.26349.007

**Figure supplement 4.** Stainings to assign molecular identity to ANS neurons based on their position, morphology and unique sensory cilia.

DOI: https://doi.org/10.7554/eLife.26349.008

**Figure supplement 5.** Stainings to assign molecular identity to ANS neurons based on their position, morphology and unique sensory cilia.

DOI: https://doi.org/10.7554/eLife.26349.009

**Figure supplement 5—source data 1.** Videos of close-up immunostainings or in situ hybridizations counterstained with acetylated tubulin antibody for neuropeptides or neuropeptide precursors expressed in the central sensory neurons SN$^{IRP2-burs}$ and SN$^{IRP2-FMRF}$.

DOI: https://doi.org/10.7554/eLife.26349.010

**Figure supplement 5—source data 2.** Videos of close-up immunostainings or in situ hybridizations counterstained with acetylated tubulin antibody for neuropeptides or neuropeptide precursors expressed in central sensory neurons SN$^{MIP1}$, SN$^{WLD}$ and SNYF$^{5cil}$.

DOI: https://doi.org/10.7554/eLife.26349.011

(*Arendt et al., 2004*) with highly extended ciliary membranes, one asymmetric neuron with five sensory cilia (SN$^{YFa5cil}$), a pair of neurons with two long parallel cilia (SN$^{WLD1}$), a pair with long branched sensory cilia (SN$^{NS29}$), and three uniciliary neurons that are part of the nuchal organ (SN$^{nuchNS}$), a putatively chemosensory annelid organ (*Purschke, 1997*). Twenty-five uniciliated neurons (23 SN$^{DLSO}$ cells, SN$^{PDF-dcl2}$, and SN$^{PDF-dcr3}$) are part of a dorsolateral sensory cluster (*Figure 1—figure supplement 1*). Most of the sensory neurons have axonal projections that are extensively branched within the neurosecretory plexus (*Figure 1—figure supplement 1*) and that are filled with dense-core vesicles (*Figure 1F*). The pairs of left-right symmetric sensory neurons project to similar areas of the neurosecretory plexus revealing a fine-scale organization within the plexus (*Figure 1G* and *Figure 1—figure supplement 1*). We refer to all neurons that project to the neurosecretory plexus as the apical nervous system (ANS).

## Comprehensive mapping of neuropeptide expression in *Platynereis* larvae

The *Platynereis* larval ANS is known to express many neuropeptides, including vasotocin, FMRFamide, and myoinhibitory peptide (MIP) (*Tessmar-Raible et al., 2007*; *Conzelmann et al., 2011*, *2013a*). To comprehensively analyze neuropeptide expression in the whole 2-day-old and 3-day-old larvae, we used whole-mount in situ hybridization for 53 and 58 *Platynereis* proneuropeptides respectively (of 98 total [*Conzelmann et al., 2013b*]). This work built upon previously published gene expression atlases for the 2-day-old and 3-day-old larval stages, which contained average registered gene expression patterns for 11 and 8 proneuropeptide genes respectively (*Conzelmann et al., 2011*, *2013a*; *Asadulina et al., 2015*). We used image registration to spatially map all proneuropeptide expression patterns to common nuclear reference templates for 2-day-old and 3-day-old

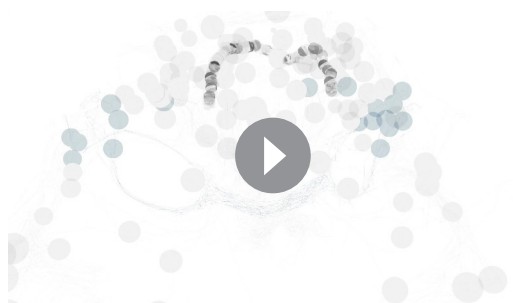

**Video 1.** Anterior view of the *Platynereis* 3 day ANS reconstructed from ssTEM data. Circles represent cell bodies and lines are axons and dendrites. ANS neurons (coloured) appear against a backdrop of neurons from the eye photoreceptor cells (dark blue-grey) and ciliated cells of the prototroch, and anterior crescent cell (light grey). The axons of the ANS neurons project to the center of the larval head, where they form a dense spherical plexus that sits anterior to the rest of the larval nervous system.

DOI: https://doi.org/10.7554/eLife.26349.013

larvae (*Asadulina et al., 2012*). We summed all binarized average expression domains and found that the region with the highest proneuropeptide expression corresponds to the ANS (*Figure 1H*). Some voxels in the gene expression atlases co-express up to 10 different proneuropeptides in both 2-day-old and 3-day-old larvae. Proneuropeptides that were expressed in the ANS region include two *Platynereis* insulin-like peptides (IRP2, IRP3), two bursicons (bursicon-A, -B), achatin, MIP, and several homologs of hypothalamic peptides (*Mirabeau and Joly, 2013*; *Jékely, 2013*) including NPY (three homologs, NPY1, NPY4, NKY2), orexin/allatotropin, tachykinin, galanin/allatostatin-A, and allatostatin-C/somatostatin (*Figure 1I* and *Figure 1—figure supplement 3*). The majority of proneuropeptides (34 genes) had very similar expression patterns between 2-day-old trochophore and 3-day-old nectochaete larval stages (*Figure 1—figure supplements 2* and *3*). Apart from this trend, a number of proneuropeptides (17 genes) showed a prominent expansion in spatial expression domain between 2-day-old and 3-day-old stages. Expansions of gene expression were mainly in the dorsolateral area of the larval head, possibly reflecting the differentiation of further peptidergic neurons in this region during development. Additionally, for six neuropeptides (CLCCY, FVMa, HFAa, NPY2, QSGa and SIFa) we could only detect expression through in situ hybridization in 3-day-old larvae.

The acetylated tubulin antibody we used to counterstain the in situ samples labels cilia and axonal scaffold. With this counterstaining signal in conjunction with wholemount in situ hybridization or immunohistochemistry in 2-day-old and 3-day-old larvae, we correlated neurons expressing specific proneuropeptides and with distinct ciliation to sensory ANS neurons reconstructed from ssTEM data (*Figure 1J*, *Figure 1—figure supplements 4* and *5*, *Video 2*). Although gene expression patterns in individual larvae had been previously published for some proneuropeptides (e.g. YFa, WLD, FMRFa (*Conzelmann et al., 2011*), RGWa (*Conzelmann and Jékely, 2012*) MIP (*Conzelmann et al., 2013b*), we repeated wholemount in situ hybridization and/or immunohistochemistry for these genes in order to obtain high resolution anterior scans to study the morphology of anterior sensory cilia. For other ANS neurons (SN$^{PDFdc}$, IN$^{RGW}$) we previously assigned neuropeptides based on direct immunogold labeling on the same EM series (*Shahidi et al., 2015*). Overall, we mapped neuropeptide expression to 26 reconstructed ANS neurons, including sensory neurons co-expressing IRP2 and FMRFamide (SN$^{IRP2-FMRF}$), IRP2 and bursicon (SN$^{IRP2-burs}$), or expressing ASTC/somatostatin (SN$^{ASTC1}$) (*Figure 1J* and *Figure 1—figure supplement 3*).

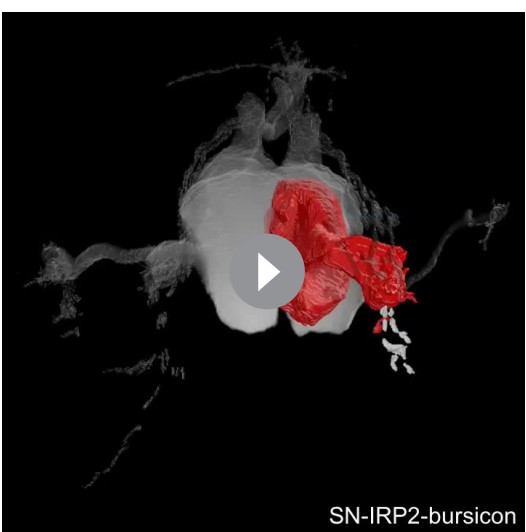

**Video 2.** Blender model of central sensory neurons of the ANS reconstructed from ssTEM data, anterior view. Only the cell body, dendrite and sensory cilia of each cell were reconstructed. Includes the SN$^{IRP2}$, SN$^{MIP1}$, SN$^{WLD}$ and SN$^{YF5cil}$ neurons. These neurons could be assigned molecular identities based on comparison of their morphology in the TEM data reconstructions to morphological data from immunostainings and in situ hybridizations counterstained with acetylated tubulin antibody (*Figure 1—figure supplements 4* and *5*).
DOI: https://doi.org/10.7554/eLife.26349.014

## Low level of synaptic connectivity within the ANS

We next analyzed how the ANS neurons are synaptically connected, based on our reconstruction of the anterior neurosecretory nervous system. We found that most neurons have no or only very few synapses (*Figure 2*). 33% of the neurons have 0–2 synapses despite highly branched axonal projections filled with dense core vesicles (*Figure 2*). This suggests that these neurons predominantly use volume transmission. The synapses we identified in most neurons contained dense-core vesicles, indicative of their peptidergic nature. The four cPRCs, an asymmetric sensory neuron (SN$^{47Ach}$), and the eight projection neurons have the highest number of synapses. In addition, 15 ANS sensory neurons have 10 or more peptidergic synapses. The cPRCs express a cholinergic marker (*Jékely et al., 2008*) and contain large synapses with clear vesicles as well as dense core vesicles. The eight projection

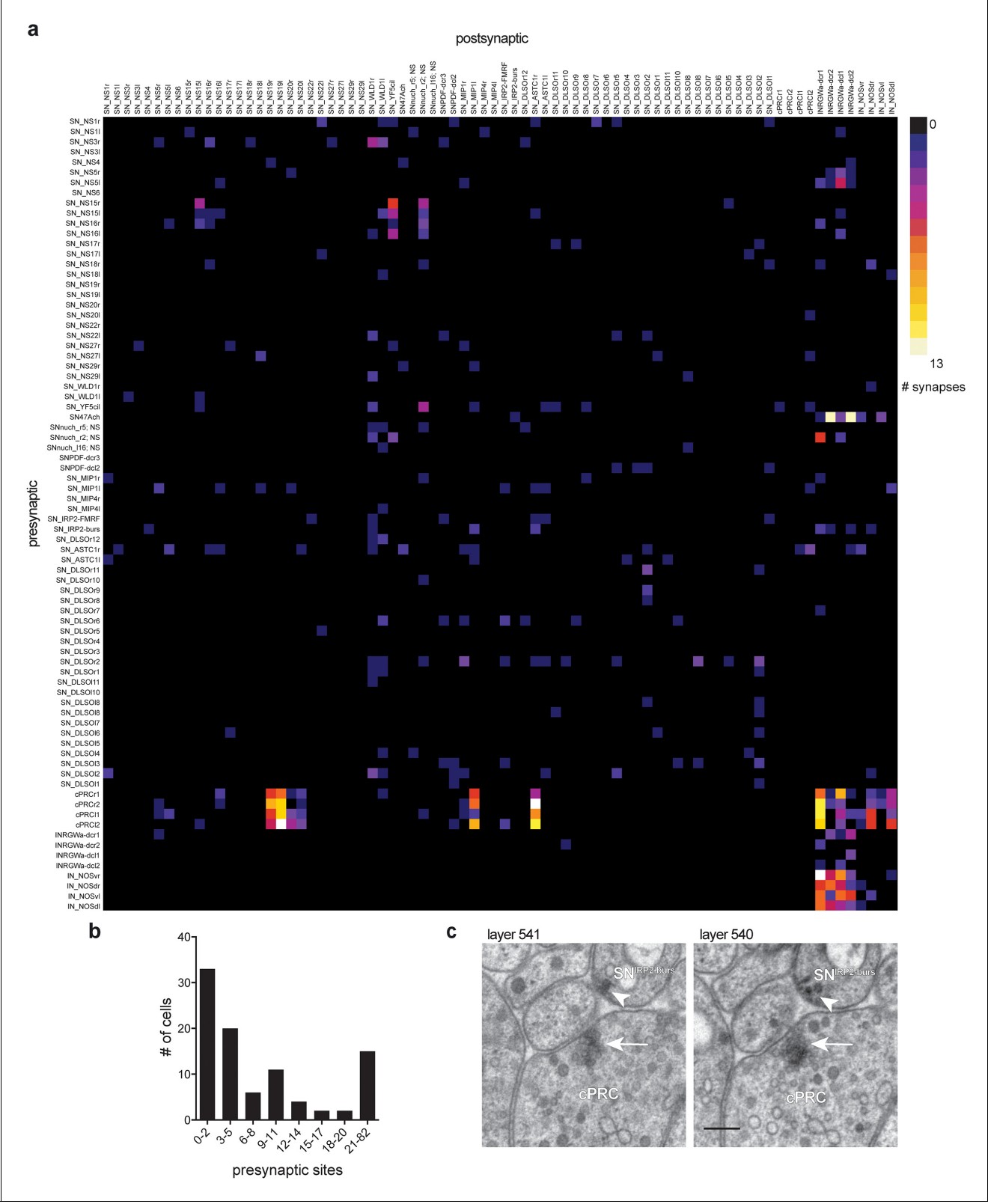

**Figure 2.** Low level of synaptic connectivity of ANS neurons. (**A**) Full matrix of synaptic connectivity of ANS neurons. With the exception of the ciliary photoreceptors (cPRC), a few sensory cells, and the IN[RGW] and IN[NOS]projection interneurons, the ANS shows very sparse synaptic connectivity. (**B**) Histogram of the number of presynaptic sites in ANS neurons. (**C**) Comparison of a cholinergic synapse of a ciliary photoreceptor (arrow) to a

*Figure 2 continued on next page*

*Figure 2 continued*

peptidergic synapse of the SN$^{IRP2-burs}$ cell (arrowhead). Two consecutive sections are shown. Scale bar in C, 200 nm. Spreadsheet of ANS synaptic connectivity data values can be found in *Supplementary file 1*.

DOI: https://doi.org/10.7554/eLife.26349.015

neurons also contain large synapses with clear vesicles and dense core vesicles. The other sensory cells have ultrastructurally distinct, small peptidergic synapses characterized by dense core vesicles clustering at the membrane (*Figure 2*).

## Analysis of peptidergic signaling networks in the ANS

The high number of neuropeptides expressed in the ANS and the low degree of synaptic connectivity of the cells prompted us to further analyze peptidergic signaling networks in the *Platynereis* larva. We used two previously published resources: an experimentally determined list of *Platynereis* neuropeptide GPCRs (*Bauknecht and Jékely, 2015*) and a spatially-mapped single-cell transcriptome dataset (*Achim et al., 2015*). The GPCR list included 18 deorphanized neuropeptide receptors. To this we added a further three deorphanized neuropeptide receptors. We deorphanized a corazonin receptor activated by both *Platynereis* corazonin-1 (CRZ1) and corazonin-2 (CRZ2), originally described as GnRH1 and GnRH2 (*Conzelmann et al., 2013a*) but renamed to follow the nomenclature in (*Zandawala et al., 2017*; *Tian et al., 2016*; *Hussain et al., 2016*), and second receptors for vasotocin and myomodulin (*Figure 3—figure supplement 1*). The single-cell transcriptome data consisted of cells of the head (episphere) of 2-day-old *Platynereis* larvae, most of which are neurons (*Achim et al., 2015*).

To comprehensively analyze potential peptide-receptor signaling networks, we first created a virtual larval brain with cells arranged in an approximate spatial map (*Figure 3A* and *Figure 3—figure supplement 2*). To this virtual map, we mapped the expression of all proneuropeptides and deorphanized receptors (*Figure 3C,D*). The combined expression of 80 proneuropeptides showed a similar pattern to the in situ hybridisation map with a highly peptidergic group of cells in the ANS region, as defined by the endocrine marker genes *Phc2*, *dimmed*, *Otp* and *nk2.1* (*Conzelmann et al., 2013a*; *Tessmar-Raible et al., 2007*) (*Figure 3B*). Given the high level of peptide expression in the ANS, the mapping of peptidergic signaling networks will mostly reveal potential signaling partners within this region or from the ANS to the rest of the brain. Most GPCR expression was also concentrated in the ANS in peptidergic cells. Several cells expressed a unique combination of up to nine deorphanized GPCRs (*Figure 3D*).

We also used the spatially mapped single cell transcriptome data to map the expression of different types of sensory receptor genes in the opsin and transient receptor potential (Trp) channel families. These sensory receptors are known for their function in the detection of light and pain/temperature/mechanical stimuli, respectively (*Terakita, 2005*; *Moran et al., 2004*). Neurons in the ANS of 2-day-old larvae express diverse combinations of these genes (*Figure 3—figure supplement 3*). This supports our conclusion from the morphological data that these neurons are responsible for detecting a variety of different environmental cues.

Co-expression analysis with small neurotransmitter synthesis enzymes revealed 1 cell (1% of total) with only neurotransmitter markers but no neuropeptide, 76 (71% of total) purely peptidergic cells, and 21 cells that co-express small transmitters and neuropeptides (19.6% of total)(*Figure 3—figure supplement 4*). The remaining 9 cells expressed neither neuropeptides nor neurotransmitter markers and thus are likely non-neuronal cells (8.4% of total). Strikingly, 2 ANS cells co-express up to 24 different proneuropeptides. Based on specific neuropeptide expression and the spatial mapping we could correlate 15 cells from the spatially mapped single cell transcriptome dataset to ANS cells reconstructed from our ssTEM data (*Supplementary file 1*). This was possible despite the two resources being derived from different larval stages, because many ANS cells are already differentiated in 2-day-old larvae and readily identifiable between stages until at least 6-day-old larvae (*Figure 3—figure supplement 5*).

To establish peptidergic signaling networks, we treated peptide-expressing cells from the spatially mapped single-cell transcriptome dataset as source nodes and GPCR-expressing cells as target nodes. We define edge weights in the directed graphs as the geometric mean of normalized

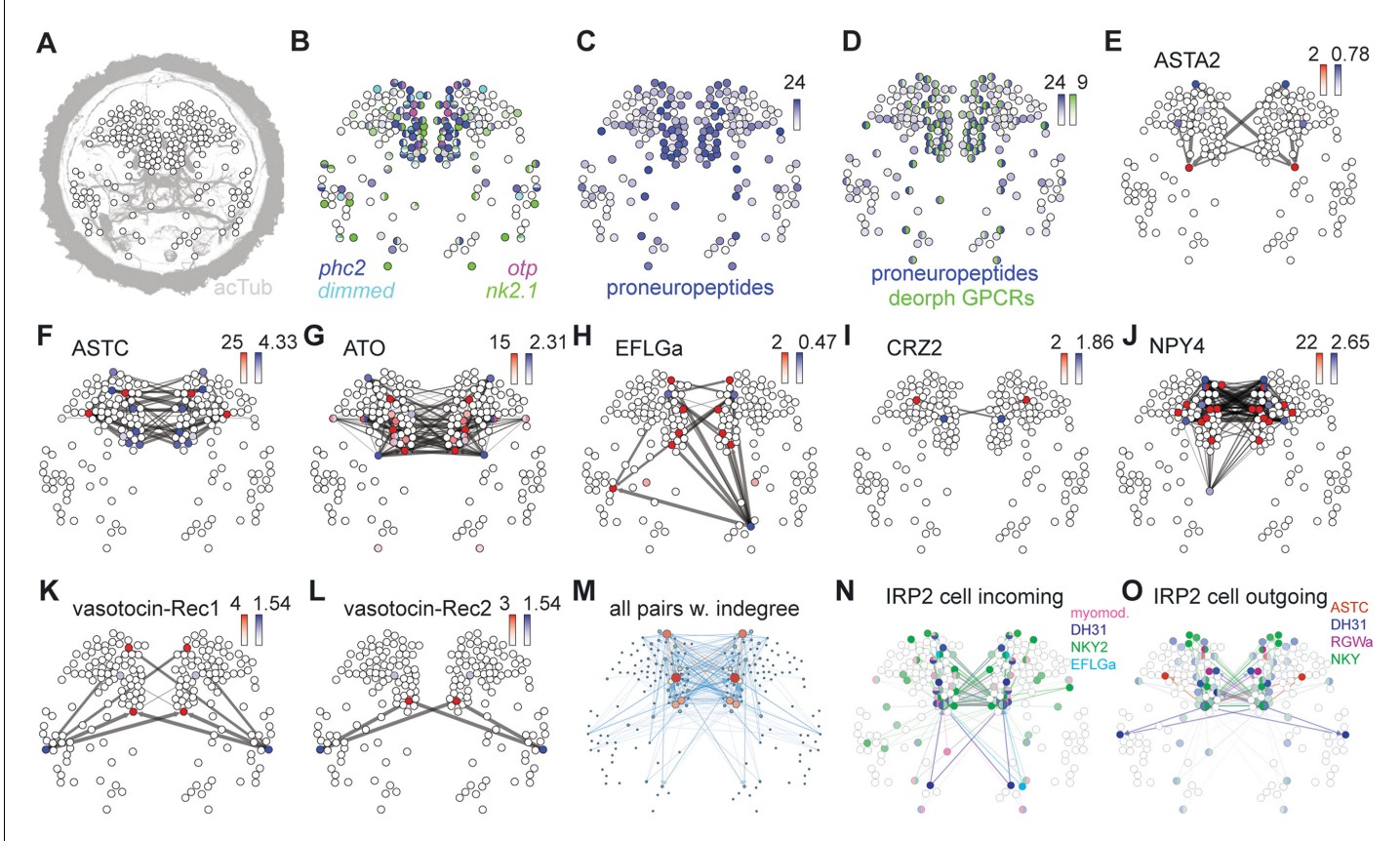

**Figure 3.** Mapping the peptidergic connectome in the *Platynereis* larval head. (**A**) Positions of cells (nodes) from single cell RNA-Seq from 2-day-old larvae placed in an approximate spatial map, projected on an acetylated tubulin immunostaining (grey, anterior view). Samples with predicted bilateral symmetry were represented as two mirror-image nodes in the map. (**B**) Expression of neuroendocrine marker genes projected on the single-cell map. Color intensity of nodes reflects magnitude of normalized $\log_{10}$ gene expression. (**C**) Map of combined expression of 80 proneuropeptides expressed in each single-cell sample. (**D**) Map of combined expression of 80 proneuropeptides and 23 deorphanized GPCRs expressed in each single-cell sample. (**E–L**) Connectivity maps of individual neuropeptide-GPCR pairs, colored by weighted in-degree (red) and proneuropeptide $\log_{10}$ normalized expression (blue). Arrows indicate direction of signaling. Arrow thickness determined by geometric mean of $\log_{10}$ normalized proneuropeptide expression of signaling cell and $\log_{10}$ normalized GPCR expression of corresponding receiving cell. (**M**) Connectivity map of all possible known neuropeptide-GPCR signaling, color and node size represent weighted in-degree. (**N**) Connectivity map of an IRP2-expressing cell with incoming neuropeptide signals. (**O**) Connectivity map of an IRP2-expressing cell with outgoing neuropeptide signals.

DOI: https://doi.org/10.7554/eLife.26349.016

The following source data and figure supplements are available for figure 3:

**Source data 1.** *Platynereis* reference transcriptome version 2.
DOI: https://doi.org/10.7554/eLife.26349.026

**Source data 2.** Normalized transformed read counts for scRNA-seq data from *Achim et al. (2015)* mapped to our *Platynereis* reference transcriptome.
DOI: https://doi.org/10.7554/eLife.26349.027

**Source data 3.** All-against-all pairwise correlation coefficients for normalized transformed read counts of scRNA-seq data from *Achim et al. (2015)* mapped to our *Platynereis* reference transcriptome.
DOI: https://doi.org/10.7554/eLife.26349.028

**Source data 4.** Gexf connectivity map files generated from scRNA-Seq data for each peptide-receptor pair, and for all peptides by all receptors.
DOI: https://doi.org/10.7554/eLife.26349.029

**Figure supplement 1.** Dose-response curves of *Platynereis* deorphanized GPCRs treated with varying concentrations of peptides.
DOI: https://doi.org/10.7554/eLife.26349.017

**Figure supplement 1—source data 1.** Raw data (luminescence measurements) from deorphanization experiments.
DOI: https://doi.org/10.7554/eLife.26349.018

**Figure supplement 2.** Spatial map of single cell RNA-Seq data from (*Achim et al., 2015*).
DOI: https://doi.org/10.7554/eLife.26349.019

**Figure supplement 3.** Expression of opsins and Trp channels in the *Platynereis* larval head.

*Figure 3 continued*

DOI: https://doi.org/10.7554/eLife.26349.020

**Figure supplement 4.** Neuropeptide-GPCR chemical connections in the *Platynereis* larval head.

DOI: https://doi.org/10.7554/eLife.26349.021

**Figure supplement 5.** Comparison of spatial distribution of sensory cilia in the anterior head of 48 hpf, 72 hpf and 6 dpf *Platynereis*.

DOI: https://doi.org/10.7554/eLife.26349.022

**Figure supplement 5—source data 1.** Tiff stacks of acetylated tubulin immunostaining from a 2-day-old, 3-day-old, and 6-day-old larva, anterior view.

DOI: https://doi.org/10.7554/eLife.26349.023

**Figure supplement 6.** Comparison of gene co-expression correlations based on sc-RNA-Seq from *Platynereis* ANS and mouse hypothalamus datasets.

DOI: https://doi.org/10.7554/eLife.26349.024

**Figure supplement 6—source data 1.** List of *Platynereis* and mouse gene orthologs 'ortholog_table_pdum_mouse_clean.txt'; Global correlation of *Platynereis* and mouse orthologous marker gene data sets 'correlation_pdum_mouse_romanov.txt', 'correlation_pdum_mouse_Campbell.txt'; Tables of correlation coefficients and p-values from comparison of *Platynereis* and mouse scRNA-Seq datasets, *Achim et al. (2015)* versus Romanov et al. 2016 'conserved_coexpression_pdum_Romanov.txt', *Achim et al. (2015)* versus *Campbell et al. (2017)*, 'conserved_coexpression_pdum_campbell.txt'.

DOI: https://doi.org/10.7554/eLife.26349.025

proneuropeptide expression in the source and normalized GPCR expression in the target. This way, we also consider the expression level of peptides and receptors. Receptor expression can correlate with neurophysiological sensitivity to a neuropeptide (*Garcia et al., 2015*; *Root et al., 2011*; *Hussain et al., 2016*). For each peptide-receptor pair, we projected these networks onto the virtual map (*Figure 3E–L* and *Figure 3—figure supplement 4*). The chemical connectivity maps of individual ligand-receptor pairs show very sparse and specific chemical wiring. On average, less than 1% of all potential connections are realized (0.8% graph density averaged for all peptide-receptor pairs). Single cells expressing many different neuropeptides generally link to non-overlapping target nodes by each peptide-receptor channel (*Figure 3O* and *Figure 3—figure supplement 4Q*). Conversely, multiple signals can converge on one cell that expresses more than one GPCR (*Figure 3N*).

The combined multichannel peptidergic connectome of 23 receptor-ligand pairs forms a single network with an average clustering coefficient of 0.49 and an average minimum path length of 1.54, forming a small-world network (*Watts and Strogatz, 1998*). Analysis of the combined network revealed highly connected components, including nodes that can act as both source and target as potential mediators of peptide cascades (*Figure 3N,O*). The three neuron types with the highest weighted in-degree and authority value include a pair of dorsal sensory neurons, a central pair of IRP2 neurons, and a pair of RGWamide-expressing neurons (*Figure 3M* and *Figure 4A*). The IRP2 neurons are under the influence of an NPY peptide and EFLGamide, the *Platynereis* homolog of thyrotropin-releasing hormone (*Bauknecht and Jékely, 2015*) and express 20 different proneuropeptides.

## Conserved molecular signature between *Platynereis* ANS and mouse hypothalamus cells

To identify similarities in gene expression between the *Platynereis* ANS and vertebrate hypothalamus, we compared the single cell transcriptome data of *Platynereis* (*Achim et al., 2015*) to a single cell transcriptome data set from a central column of the mouse hypothalamus, consisting of 3131 cells (*Romanov et al., 2017*), and a single cell transcriptome data set of 13,079 neurons from the mouse hypothalamic arcuate-median eminence complex (*Campbell et al., 2017*). By comparing the expression across cells of orthologous neuropeptides, neuropeptide receptors, transcription factors and other marker genes in these data sets (*Figure 3—figure supplement 6*), we identified significant coexpression for a subset of genes in both mouse and *Platynereis* marking a cell population with a likely function in regulating circadian oscillations. In all three datasets the genes *bmal/arntl*, *otp*, *synaptotagmin/syt5*, and *7B2/SCG5* showed significant co-expression. In both the Romanov et al. mouse data and the *Platynereis* data, these genes are also co-expressed with *ASTA2/galanin* neuropeptide. The *bmal* gene encodes an important component of the circadian clock in flies and vertebrates (*Van Gelder et al., 2003*) and is expressed in the *Platynereis* ANS, in the same region as several other circadian clock markers (*Arendt et al., 2004*; *Tosches et al., 2014*). Co-expression correlation analyses of the two datasets also revealed co-expression in all three datasets of the neuropeptide pair *ASTC/somatostatin* and *sulfakinin/cholecystokinin*. Additionally, neuropeptide pairs

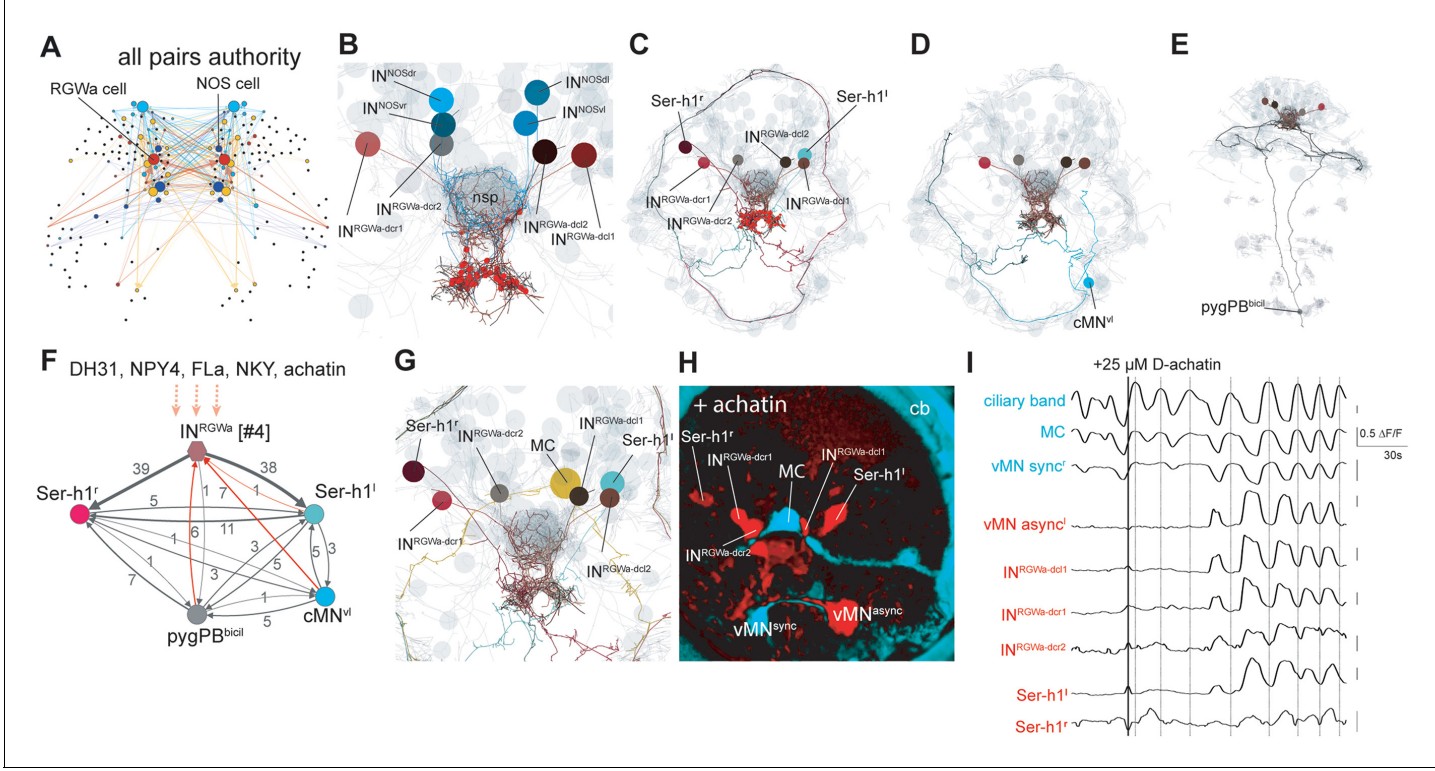

**Figure 4.** Projection neurons connect the ANS to the synaptic nervous system. (A) Connectivity map of all possible known neuropeptide-GPCR signaling, colors represent modules defined by randomized community detection analysis in Gephi and node size represents authority. (B) Reconstructed IN[RGWa] and IN[NOS] projection neurons (ANS) (shades of brown and cyan) in a full body transmission electron microscopy (ssTEM) dataset of a 3-day-old larva, shown against a framework of ANS neurons (light grey). Axons and dendrites appear as lines and cell body positions are represented by spheres. (C) ssTEM reconstruction of IN[RGWa] projection neurons and Ser-h1 serotonergic neurons. Red spheres indicate IN[RGWa] presynaptic sites. (D) ssTEM reconstruction of IN[RGWa] projection neurons and a presynaptic ciliomotor neuron, cMN[vl]. (E) ssTEM reconstruction of IN[RGWa] neurons and a presynaptic sensory neuron, pygPD[bicil]. (F) Synaptic connectivity graph of IN[RGWa], Ser-h1, pygPD[bicil], and cMN[vl] neurons. Synaptic inputs from the synaptic nervous system to the IN[RGWa] neuron are in red. Peptidergic inputs to the IN[RGWa] neurons are indicated with dashed arrows. (G) ssTEM reconstruction of IN[RGWa] projection neurons, Ser-h1 neurons, and the cholinergic ciliomotor MC neuron, anterior view (H) Confocal microscopy image of correlated pixels of GCaMP6s signal in a 2-day-old *Platynereis* larva after the addition of 25 µM D-achatin neuropeptide, anterior view. Cells showing correlated activity with the serotonergic neurons (red) and the MC cell (cyan) are shown. IN[RGWa-dcl2] could not be identified in this larva and is likely obscured by the MC cell, Ser-h1[l] and/or IN[RGWa-dcl1]. (I) Neuronal activity patterns of individually identified neurons in a 2-day-old larva treated with 25 µM achatin.

DOI: https://doi.org/10.7554/eLife.26349.030

The following source data and figure supplements are available for figure 4:

**Source data 1.** Video of calcium imaging in a *Platynereis* larva used to calculate neuronal activity correlations in *Figure 4H* and neuronal activity patterns in *Figure 4I* (.tiff file).
DOI: https://doi.org/10.7554/eLife.26349.034

**Figure supplement 1.** Spatial distribution of dense core vesicles and synapses in an RGWa-expressing projection interneuron, IN[RGW-dcr1].
DOI: https://doi.org/10.7554/eLife.26349.031

**Figure supplement 2.** (A–D) Standard deviation in fluorescent calcium signal of the (A) Ser-h1[l], (B) Ser-h1[r], (C) IN[RGWa-dcl1] and (D) IN[RGWa-dcr1] neurons in 2-day-old larvae before and after treatment with 25 µM D-achatin.
DOI: https://doi.org/10.7554/eLife.26349.032

**Figure supplement 2—source data 1.** Correlation values of neuronal activity patterns and standard deviation of GCaMP6 fluorescence in achatin-treated larvae.
DOI: https://doi.org/10.7554/eLife.26349.033

*ASTA2/galanin* and *glycoprotein beta/Gpha2*, and *NPY* and *ASTC/somatostatin* showed co-expression in both the Romanov et al. mouse and the *Platynereis* datasets. These co-expressions may indicate neuropeptides involved in conserved combinatorial signaling in both the mouse and *Platynereis* neuroendocrine systems.

## Projection interneurons connect the peptidergic ANS to the synaptic nervous system

Based on the spatially mapped *Platynereis* single cell transcriptome dataset, the RGWamide-expressing neurons express *VAChT*, two NPY receptors (NPY4 and NKY), an achatin receptor, and 17 proneuropeptide genes. By position and RGWamide-expression we identified these cells in our TEM reconstruction as IN[RGW] projection neurons. We also identified cells in the single cell transcriptome data that likely correspond to IN[NOS] projection neurons in our ssTEM reconstruction, based on RYamide expression and expression of *nitric oxide synthase* (unpublished results). The four IN[RGW] projection interneurons as well as the four IN[NOS] projection interneurons have a distinct anatomy (*Figure 4B*). They lack sensory dendrites or cilia and their axons are extensively branched, spreading around the circumference of the ANS neuropil and projecting posteriorly out of this area to the ventral neuropil of the synaptic nervous system. Here the IN[RGW] cells form many large (likely cholinergic) synapses with clear synaptic vesicles on two serotonergic ciliomotor neurons (Ser-h1)(*Figure 4C*) that are known to play a role in the regulation of ciliary beating (*Verasztó et al., 2017*). The axons of the IN[RGW] cells are also filled with dense core vesicles throughout the region where they innervate the neuropil (*Figure 4—figure supplement 1*, *Video 3*). These vesicles accumulate in bulbous thickenings of the IN[RGW] neurites, however no peptidergic synapses were evident in these cells. This indicates that the IN[RGW] cells can signal either synaptically with neurotransmitters or via peptidergic endocrine release. The IN[NOS] cells only synapse on the IN[RGW] cells in the ventral neuropil. The IN[NOS] cells synapse on the IN[RGW] cells with mixed clear and dense core vesicle synapses (data not shown). The IN[RGW] neurons also receive synapses in the ventral neuropil area from two non-ANS neurons. These neurons (cMN[vl] and pygPB[bicil]) belong to the ciliomotor circuit of the larva (*Figure 3D–F*). From the ANS, the IN[RGW] projection neurons receive small transmitter synapses from the cPRC cells, mixed clear and dense core vesicle synapses from the SN[47Ach] neuron in the ventral neuropil, and peptidergic synapses from three other ANS sensory neurons in the neurosecretory plexus (*Figure 4*). The distinct anatomy and connectivity as well as the high authority values of the projection neurons in the peptidergic network indicate that these cells are important in relaying peptidergic signals from the ANS to the rest of the brain.

To test how the projection interneurons respond to neuropeptides, we used calcium-imaging experiments. The IN[RGW] and IN[NOS] neurons express seven deorphanized GPCRs (*Figure 4F*), including a receptor for achatin that is specifically activated by achatin peptide containing a D-amino acid (*Bauknecht and Jékely, 2015*). When we treated larvae with D-achatin, neurons in the ANS were rhythmically activated. These neurons correspond by position to the IN[RGW] cells (*Figure 4I*). The rhythmic activation was in-phase with the activation of the Ser-h1 neurons and a ventral motorneuron (vMN[async]), but out of phase with the main cholinergic motorneuron of the head ciliary band, the MC neuron (readily identifiable by calcium imaging [*Verasztó et al., 2017*]). We could not observe a similar effect upon L-achatin treatment (data not shown). *Platynereis* larvae have a cholinergic and a serotonergic ciliomotor circuit that oscillate out-of-phase. The cholinergic phase arrests the cilia and the serotonergic phase correlates with ciliary beating (*Verasztó et al., 2017*). Correlation analysis of neuronal activity patterns revealed that D-achatin coupled the activity of the IN[RGW] cells to the serotonergic cells, and increased the negative correlation between the activity patterns of serotonergic neurons and the MC neuron (*Figure 4I* and *Figure 4—figure*

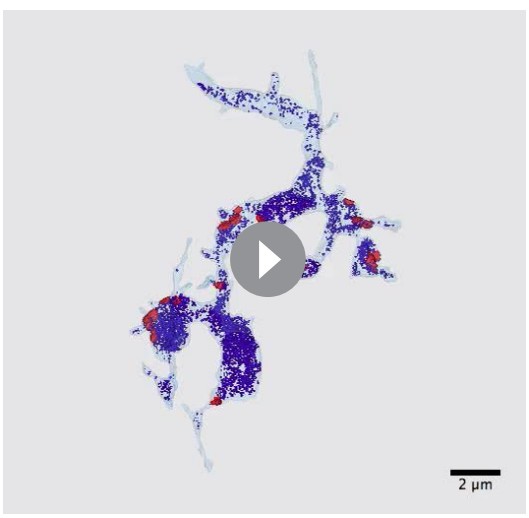

**Video 3.** 3D reconstruction from 50 TEM sections of a segment of the axon of the projection interneuron IN[RGWa-dcr1], to show distribution of dense core vesicles and classical synapses. The axon is coloured light blue, synaptic sites are red, and dense core vesicles purple. Scale bar = 2 µm.
DOI: https://doi.org/10.7554/eLife.26349.035

*supplement 2*). This provides an example of how peptidergic signaling in the ANS could recruit neurons to a rhythmically active circuit, enhance the rhythm, and thereby potentially influence locomotor activity.

## Discussion

Here we presented a comprehensive anatomical description of the ANS in the *Platynereis* larva. We combined this with the cellular-resolution mapping of neuropeptide signaling components to analyze potential peptidergic signaling networks. The use of scRNA-seq has a great potential to reveal such signaling networks but also has limitations. For example, we could only score proneuropeptide and receptor mRNA expression and not protein expression levels, peptide release, or the degree of neuronal activation. We also only analyzed a relatively small single-cell dataset derived from the head of the larva and thus could not investigate long-range neuroendocrine signaling from the ANS to the rest of the body. The first analyses we present here can be extended when more data become available and can also be applied to other species (e.g., [*Campbell et al., 2017*; Romanov et al. 2017]). Nevertheless, our approach can reveal all potential peptidergic connections and allows the identification of target cells for different neuropeptide signaling channels, and the development of hypotheses on peptidergic signaling that can be experimentally tested, as we have shown here for achatin.

We found that the *Platynereis* larval ANS has a low degree of synaptic connectivity, a strong neurosecretory character, and a high diversity of neuropeptide expression. This suggests that the ANS is wired primarily chemically and not synaptically. We identified very specific peptidergic links based on proneuropeptide and GPCR transcript expression that connect only a small subset of ANS neurons. Strikingly, it is not only the proneuropeptides that are expressed with high diversity in the ANS, but also their receptors, suggesting that most peptidergic signaling in the brain occurs within the ANS. Through our analysis of peptidergic signaling networks in the ANS, we could also find evidence that the ANS undergoes autoregulation, at the level of single cells or cellular subpopulations. In *Platynereis*, autoregulation of cells in the ANS can occur through allatotropin/orexin, NPY4, achatin, diuretic hormone, excitatory peptide, FMRFa, MIP or myomodulin signaling. These are cells where co-expression of the neuropeptide and its corresponding deorphanized receptor occurs in the same cell, based on the single cell RNA-Seq data analysis. A high diversity of neuropeptide receptor expression has also been reported for the mouse hypothalamus (*Campbell et al., 2017*). These observations suggest that there is more extensive peptidergic integration within these centers than was previously appreciated. Neurosecretory centers may thus also function as chemical brains wired by neuropeptide signaling where the specificity is derived not from direct synaptic connections but by peptide-receptor matching. Given a high enough number of specific peptide-receptor pairs and allowing the possibility of combinatorial signaling, it is theoretically possible to wire arbitrarily complex neural networks. We interpret these results as an indication that complex signaling is already occurring between the cells of the ANS before any endocrine signals are released to the other regions of the body. This extends the concept of neurosecretory centers as organs that release peptides to influence external downstream targets.

Previously, it was found that the *Platynereis* larval ANS expressed the neuropeptides FMRFa and vasotocin, suggesting this region has homology to the vertebrate hypothalamus (*Tessmar-Raible et al., 2007*; *Tessmar-Raible, 2007*). Through the analyses carried out here, we could assign orthologs of additional neuropeptides known to be involved in hypothalamic signaling, including NPY, orexin, tachykinin, galanin and somatostatin, to the *Platynereis* ANS. By comparing the *Platynereis* ANS scRNA-seq data to scRNA-seq data from mouse hypothalamus, we could also identify shared molecular fingerprints relating to circadian rhythmicity. These molecular similarities strengthen the argument for an evolutionary relationship between the ANS and the hypothalamus.

The *Platynereis* ANS incorporates the ciliary photoreceptor cells, previously postulated to function in non-visual light detection for regulating circadian cycles in the free-swimming larvae (Tosches et al. 2014). The ANS also includes a subset of cells associated with the *Platynereis* larval apical sensory organ (*Marlow et al., 2014*). These are namely the central-most cells of the ANS, such as SN$^{WLD}$, SN$^{IRP2}$, and SN$^{MIP}$, implicated to play a role in larval settlement (*Marlow et al., 2014*; *Conzelmann et al., 2013a*). Not all cells of the apical organ as described by Marlow et al. project to the ANS neuropil, however – the mechanosensory cells bypass this neurosecretory center to synapse

on a separate synaptic circuit (unpublished results). In many marine invertebrate larvae, the apical sensory organ is lost following settlement and metamorphosis (Nielsen 2005). We found that while the central ciliated crescent cell is lost in 6-day-old settled postlarvae, the sensory cells SN$^{WLD}$, SN$^{MIP}$, and SN$^{IRP2}$ remain and may continue to play a sensory-neurosecretory role later in the life cycle. In *Platynereis*, parts of the larval nervous system are thought to persist in the adult nervous system, however exactly which cells remain into adulthood beyond the 6-day-old postlarval stage has not yet been described at the level of specific individual cells. Certainly the sensory systems of the ciliary photoreceptors and nuchal organs, which project to the ANS neuropil, remain into adulthood (*Orrhage and Müller, 2005*; *Purschke, 2005*; *Zantke et al., 2013*). This suggests that the ANS, which initially integrates environmental cues for regulating larval swimming and settlement behavior in free-swimming, non-feeding *Platynereis* larvae, will later integrate environmental cues in order to regulate the timing of feeding and other cyclic behavioral or physiological activities, likely including the maintenance of circadian rhythm throughout the life cycle (Zantke et al. 2013). The neuropeptides investigated here also continue to be expressed during the adult stages of the *Platynereis* life cycle (*Conzelmann et al., 2013a*). Neuropeptide function in *Platynereis* adult stages has not yet been characterized, however, in other annelid groups - leeches and earthworms - vasopressin/oxytocin family peptide hormones induce adult reproductive behavior or egg-laying behavior respectively (*Oumi et al., 1996*; *Wagenaar et al., 2010*).

The ANS, a primarily neurosecretory anterior brain region, is directly sensory in *Platynereis*, possibly reflecting an ancestral condition for neurosecretory centers (*Tessmar-Raible et al., 2007*; *Hartenstein, 2006*). An anatomically and molecularly distinct 'chemical' and 'synaptic' brain as we found in the *Platynereis* larva may have originated early in the evolution of the nervous system. This idea is consistent with the chimeric brain hypothesis, which posits the fusion of a neurosecretory and a synaptic nervous system early in animal evolution (*Tosches and Arendt, 2013*) and is also supported by the distinct molecular signature of ANS cells (*Achim et al., 2017*). Interestingly, the projection interneurons and ciliary photoreceptor cells exhibit features of both the chemical (dense core vesicles, neuropeptide and GPCR expression) and synaptic (large synapses with clear vesicles, neurotransmitter marker expression) nervous system cell types. The dual peptidergic-synaptic nature of the projection interneurons, which are themselves under peptidergic control, enables them to act as translators of environmental cues integrated through the ANS into locomotor outputs in the synaptic nervous system. Specifically, the IN$^{RGW}$ projection interneurons link the sensory-neurosecretory ANS to the serotonergic ciliomotor neurons through synaptic connections. This would allow regulation of ciliary swimming, a larva-specific feature.

Neuropeptides are ideal molecules for chemical signaling due to their small size and high diversity. Many of the neuropeptides expressed in the *Platynereis* ANS and the vertebrate hypothalamus belong to ancient peptide families that evolved close to the origin of the nervous system (*Jékely, 2013*; *Mirabeau and Joly, 2013*). Neuropeptide genes have been identified in bilaterians, cnidarians, and even the early-branching metazoan *Trichoplax adhaerens*, that lacks a synaptic nervous system but has neurosecretory cells (*Nikitin, 2015*; *Smith et al., 2014*). Chemical signaling by neuropeptides may have been important early during nervous system evolution in small metazoans where only peptide diffusion could coordinate physiological activities and behavior (*Varoqueaux and Fasshauer, 2017*; *Senatore et al., 2017*). As metazoans grew larger, the coordination of their large complex bodies required control by a synaptic nervous system (*Keijzer and Arnellos, 2017*). The separate origins of the chemical and the synaptic nervous system may still be reflected to varying degrees in contemporary brains. The *Platynereis* larval nervous system, and possibly the nervous system of other marine invertebrate larvae, shows a particularly clear segregation of non-synaptic and synaptic nervous systems. The extent of non-synaptic signaling, even in the relatively simple nervous system of the *Platynereis* larva, highlights the importance of the combined study of connectomes and chemical signaling networks.

## Materials and methods

### Electron microscopy reconstruction of the anterior neurosecretory plexus in *Platynereis*

We reconstructed the circuitry of cells in the anterior plexus from a 3-day-old *Platynereis* ssTEM dataset we generated previously (*Randel et al., 2015*). The dataset consists of 4845 layers of 40 nm thin sections. Preparation, imaging, montage, and alignment of the dataset is described (*Randel et al., 2015*). The cells were reconstructed, reviewed, 3D visualized, and the resulting synaptic network was analyzed in Catmaid (*Saalfeld et al., 2009*; *Schneider-Mizell et al., 2016*).

Morphology and distribution of dense core vesicles and synapses in the $IN^{RGWa-dcr1}$ projection interneuron was reconstructed using TrakEM2 (*Cardona et al., 2010*; *Cardona et al., 2012*). For comparing the morphology of cells from in situ hybridization and immunohistochemistry scans to cells reconstructed from the TEM data, morphology of the cell body, dendrites and cilia of the sensory neurons $SN^{IRP2}$, $SN^{MIP1}$, $SN^{WLD}$ and $SN^{YF5cil}$ was reconstructed using TrakEM2. TrakEM2 reconstructions were exported as wavefront files and imported into Blender 2.7.1 (http://www.blender.org/) for the generation of 3D videos.

### Neuropeptide Expression Atlas in 2-day-old and 3-day-old *Platynereis*

DIG-labeled antisense RNA probes were synthesized from clones sourced from a *Platynereis* directionally cloned cDNA library in pCMV-Sport6 vector (*Conzelmann et al., 2013a*) or PCR amplified and cloned into the vectors pCR-BluntII-TOPO or pCRII-TOPO. Larvae were fixed in 4% paraformaldehyde (PFA) in 1 X PBS with 0.1% Tween-20 for 1 hr at room temperature. RNA in situ hybridization using nitroblue tetrazolium (NBT)/5-bromo-4-chloro-3-indolyl phosphate (BCIP) staining combined with mouse anti-acetylated-tubulin staining, followed by imaging with a Zeiss LSM 780 NLO confocal system and Zeiss ZEN2011 Grey software on an AxioObserver inverted microscope, was performed as previously described (*Asadulina et al., 2012*), with the following modification: fluorescence (instead of reflection) from the RNA in situ hybridization signal was detected using excitation at 633 nm in combination with a Long Pass 757 filter. Animals were imaged with a Plan-Apochromat 40x/1.3 Oil DIC objective.

We registered thresholded average gene expression patterns of >5 individuals per gene to common 2-day-old and 3-day-old whole-body nuclear reference templates generated from DAPI nuclear staining. New reference templates were generated as previously described (*Asadulina et al., 2012*), using a combination of individual larva scans contributed by the Jékely lab and the lab of Detlev Arendt, European Molecular Biology Laboratory (EMBL), Heidelberg, Germany. The 2-day-old nuclear reference template was generated from scans of 145 larvae, and the 3-day-old template was generated from scans of 139 larvae. Acetylated tubulin counterstaining from each larva was also registered to the nuclear reference template, then averaged to generate a morphological framework of cilia and axonal tracts upon which to project gene expression patterns. The acetylated tubulin average was generated from 94 and 75 scans for the 2- and 3-day-old larval stages, respectively. Gene expression atlases were set up in the visualization software Blender (https://www.blender.org/) as described (*Asadulina et al., 2015*).

For comparing the morphology of cells from in situ hybridization scans to cells reconstructed from the ssTEM data, we carried out high-resolution scans of the anterior larval head. Snapshots and 3D videos of these cells were constructed using Imaris 7.6.3 software.

### Deorphanization of receptor-peptide pairs

*Platynereis* GPCRs were identified in a reference transcriptome assembled from cDNA generated from 13 different life cycle stages (*Conzelmann et al., 2013b*). GPCRs were cloned into pcDNA3.1 (+) (Thermo Fisher Scientific, Waltham, USA) and deorphanization assays were carried out as previously described (*Bauknecht and Jékely, 2015*). We carried out deorphanization assays for corazonin receptor (ATG31107), myomodulin receptor 2 (ATG31108), and vasotocin receptor 2 (ATG31106).

### Analysis of single cell transcriptome data

Fastq files containing raw paired-end RNA-Seq data for 107 single cells from the 48 hr old *Platynereis* larval episphere (*Achim et al., 2015*) were downloaded from ArrayExpress, accession number

E-MTAB-2865 (https://www.ebi.ac.uk/arrayexpress/experiments/E-MTAB-2865/). Only those samples annotated as 'single cell' and with a corresponding spatial mapping prediction by *Achim et al. (2015)* were used in our analysis.

Fastq files with raw paired-end read data were loaded into CLC Genomics Workbench v6.0.4 (CLC Bio). Data were filtered to remove Illumina adapter primer sequences, low quality sequence s (Quality Limit 0.05) and short fragments (less than 30 base pairs). Filtered data were mapped to the assembled *Platynereis* reference transcriptome including only sequences with a BLASTX hit e-value <1e-5 to the SwissProt database, plus all previously described *Platynereis* genes, including 99 proneuropeptides, a total of 52,631 transcripts. Mapping was carried out in CLC Genomics Workbench v6.0.4 using the RNA-Seq Analysis function, with the following mapping parameters: paired distance 100–800 base pairs; minimum length fraction 0.8, minimum similarity fraction 0.9, maximum number of mismatches 2. The total number of reads mapped to each gene, normalized by gene length (reads per kilobase million (RPKM)) in each sample was assembled into one spreadsheet using the 'Experiment' function, and this spreadsheet was exported as an Excel file. RPKM data for each gene and sample were converted in Excel to counts per million (cpm) by dividing RPKM by the total number of mapped reads in each sample and multiplying by $10^6$ followed by conversion to a log base 10.

Single cell samples were sourced from populations consisting of dissociated cells of several individual larvae (*Achim et al., 2015*), therefore some of the RNA-Seq samples could represent sequencing of the same cell from different larvae. To determine a cutoff for transcriptome similarity to merge samples representing the same cell from different individuals, we calculated the all-against-all pairwise correlation coefficients. Plotting these correlation coefficients as a histogram showed a prominent peak of highly correlated samples. We interpret these as samples deriving from the re-isolation of the same cell from different larvae. We used a cutoff of 0.95 Pearson correlation as a cutoff, above which we merged samples as representing the same cell by using the mean normalized expression value for each gene.

Sample names were imported as nodes into software for graph visualization and manipulation, Gephi.0.8 beta (http://gephi.org). Nodes were manually placed in position in a Gephi map in an approximate 2D representation of the 3D spatial predictions of each cell generated by Achim et al. (*Achim et al., 2015*) based on a wholemount in situ hybridization gene expression atlas of 72 genes (https://www.ebi.ac.uk/~jbpettit/map_viewer/?dataset=examples/coord_full.csv&cluster0=examples/resultsBio.csv). Samples with predicted bilateral symmetry were represented as two mirror image nodes in the map (left and right), while samples with predicted asymmetry were represented as single nodes. Node position coordinates for each sample were saved and exported as gexf connectivity file for use in generating virtual gene expression patterns and peptide-receptor connectivity maps.

Virtual expression patterns for each gene were generated by converting normalized log10 gene expression values into node color intensity RGB values in the Gephi map. Connectivity files for each peptide-receptor pair were generated by preparation of connectivity data files as csv files where the geometric mean of normalized log10 peptide expression from the 'sending cell' and corresponding GPCR expression from the 'receiving cell' was used as a proxy for connectivity strength. These connectivity data csv files were imported into Gephi to generate gexf connectivity maps, and random node positions were replaced with node position coordinates for each cell from the spatial position gexf map described above. An 'all-by-all' connectivity map representing the potential cellular signaling generated by all known peptide-receptor pairs was generated by adding the connectivity data from each peptide-receptor pair into a single connectivity file.

Connectivity maps were analyzed for degree of connectivity, graph density, modularity, number of connected components, and clustering coefficient in Gephi.0.8 beta. Nodes in connectivity maps were colored by weighted in-degree (average number of incoming edges per node, adjusted for connectivity strength). Edge thickness was used to represent connectivity strength (geometric mean of peptide X receptor normalized gene expression). Authority was assigned to nodes using a Hyper-link-Induced Topic Search (HITS). Higher authority values indicate nodes that are linked to greater numbers of other nodes, or in the case of our peptidergic signaling maps, neurons with the capacity to receive signals from and send signals to the greatest number of other cells. Following export from gephi as svg files, corresponding virtual neuropeptide expression for each connectivity map was used to color signaling cells by overlaying in Adobe Illustrator CS6 V6.0.0 (Adobe Systems Inc.)

## Comparison of mouse and *Platynereis* single-cell transcriptomes

For the comparison of *Platynereis* ANS and mouse hypothalamus cells, mouse hypothalamus single cell transcriptomic counts were downloaded from the Gene Omnibus Database (https://www.ncbi.nlm.nih.gov/geo/query/acc.cgi?acc=GSE74672 for the 'Romanov et al.' dataset, file GSE74672_expressed_mols_with_classes.xlsx.gz file, 24341 genes and 2881 hypothalamus cells) https://www.ncbi.nlm.nih.gov/geo/query/acc.cgi?acc=GSE93374 for the 'Campbell et al.' dataset, file GSE93374_Merged_all_020816_DGE.txt.gz, 26774 genes and 13079 neuronal cells).

We established a list of 84 informative *Platynereis* cell type marker genes from the literature, and used BLAST searches (e-value cutoff 1e-10) to derive a list of orthologous marker genes between *Platynereis* and mouse (*Figure 3—figure supplement 6—source data 1*). For large gene families, we constructed phylogenetic trees. Some of the orthology assignments were based on the literature. Each *Platynereis* marker gene had one or more orthologous genes in mouse. For each orthologous gene pair, the Pearson correlation was computed using the count values of all *Platynereis* and mouse cells. When there were several mouse genes associated with a *Platynereis* gene, only the gene associated with the lowest p-value of a correlation test between the *Platynereis* and mouse genes was retained. The correlation test p-values were generated with the R function rcor.test with default parameters. To control for multiple testing on each of the gene-to-gene correlations tested and to assess the significance of the global correlation between *Platynereis* and the two mouse datasets on the orthologous genes, we used a 1000X randomization scheme. This scheme was used both to infer a significance value for the test of global correlation between orthologous genes, and to correct for multiple testing of correlations between gene pairs.

Specifically, the 1000 randomised genes X cells matrix of transcript counts were generated as follows. For every gene, a new permutation was made on the cells. For that gene each cell was reassigned an expression value that corresponded to the value of that gene from a different cell, following a random permutation that was recomputed for that gene. This was done 1000 times, both for the *Platynereis* data and the two mouse datasets. Then, using the orthology table, we computed a global *Platynereis*-mouse correlation on the correlation of the marker genes, excluding values corresponding to the correlation between identical markers. Most of the excluded values lie on the diagonal and are one in both the *Platynereis* and the mouse datasets, and would induce a spurious correlation if they were taken into account in the calculation of the global correlation. This global correlation was statistically significant according to both the R rcor.test (correlation = 0.11, p-value=3.2e-11, 'Romanov et al.' dataset, correlation = 0.12, p-value=2.3e-13, 'Campbell et al.' dataset) and the p-value generated through the 1000X randomized scheme (p-value<0.001, '*Romanov et al., 2017*' dataset, p-value<0.001, '*Campbell et al. (2017)*' dataset) (*Figure 3—figure supplement 6D, E*).

We listed the pairs of orthologous genes that were found in our data to be significantly co-expressed in both species (*Figure 3—figure supplement 6—source data 1*). For this we needed to correct for multiple testing of correlations. We re-used the 1000X randomized data to derive a distribution of p-values for each of the correlation tests and based on the comparison between the actual correlation test p-values and the simulated data-based p-values we derived corrected p-values accounting for the multiple testing.

For all three datasets, correlations between genes were visualized using the R package corrplot in which we masked the correlation where the corrected p-value was >0.05. The order of the orthologous marker genes was kept identical between the *Platynereis* and mouse datasets to enable a direct visual comparison of the correlation patterns. In addition, for every gene pair, a Fisher's exact test was computed using the R function fisher.test on the presence/absence of expression of a given gene in cells, and the Maximal Information Coefficient (MIC, [*Reshef et al., 2011*]) was calculated to provide information on the possible non-linear relationship between the genes. These are provided in the summary table of significant marker gene associations (*Figure 3—figure supplement 6—source data 1*).

The calculations were done in the R package with a custom script (*Williams et al., 2017*, copy archived at https://github.com/elifesciences-publications/Williams_et_al_2017).

## Immunohistochemistry

Wholemount triple immunostaining of 2 and 3 day old *Platynereis* larvae fixed with 4% paraformaldehyde were carried out using primary antibodies raised against RGWa neuropeptide in rat (CRGWa) and achatin neuropeptide in rabbit (CGFGD), plus a commercial antibody raised against acetylated tubulin in mouse (Sigma T7451). Double immunostaining was carried out with primary antibodies raised against MIP (*Conzelmann et al., 2011*), MLD, FMRFa (*Shahidi et al., 2015*), or YFa neuropeptides raised in rabbit and commercial acetylated tubulin antibody raised in mouse. The synthetic neuropeptides contained an N-terminal Cys that was used for coupling during purification. Antibodies were affinity purified from sera as previously described (*Conzelmann and Jékely, 2012*). Immunostainings were carried out as previously described (*Conzelmann and Jékely, 2012*).

For comparing the morphology of cells from immunohistochemistry scans to cells reconstructed from the TEM data we carried out high-resolution scans of the anterior larval head using a Leica TCS SP8 confocal system with an HC PL APO 40 × 1.30 OIL CS2 objective and LAS X software. Snapshots and 3D videos of these cells were constructed using Imaris 7.6.3 software.

## Calcium imaging experiments

Fertilized eggs were injected as previously described (*Conzelmann et al., 2013b*) with capped and polA-tailed GCaMP6 RNA generated from a vector (pUC57-T7-RPP2-GCaMP6s) containing the GCaMP6 ORF fused to a 169 base pair 5′ UTR from the *Platynereis* 60S acidic ribosomal protein P2. The injected individuals were kept at 18°C until 2-days-old in 6-well-plates (Nunc multidish no. 150239, Thermo Scientific). Calcium imaging was carried out on a Leica TCS SP8 upright confocal laser scanning microscope with a HC PL APO 40x/1.10 W Corr CS2 objective and LAS X software (Leica Microsystems) GCaMP6 signal was imaged using a 488 nm diode laser at 0.5–4% intensity with a HyD detector in counting mode. 2-day-old *Platynereis* larvae were immobilized for imaging by gently holding them between a glass microscope slide and a coverslip raised with 2 layers of tape as spacer. Larvae were mounted in 10 μl sterile seawater. For peptide treatment experiments, individual larvae were imaged for 2 min in the plane of the RGWamide interneurons and MC cell to assess the state of the larval nervous system prior to peptide treatment. For treatment, larvae were imaged for a further 5.5 min. 10 μl 50 μM synthetic D-achatin (G{dF}GD) (final concentration 25 μM) dissolved in seawater was added to the slide at the 1 min mark by slowly dripping it into the slide-coverslip boundary, where it was sucked under the coverslip. As a negative control, larvae were treated with synthetic L-achatin (GFGD) at the same concentration. Receptor deorphanization experiments have shown that the D-form of achatin activates the achatin receptor GPCR, whereas the L-form of achatin does not (*Bauknecht and Jékely, 2015*). D-achatin response was recorded from 12 larvae, and L-achatin response was recorded from six larvae. Calcium imaging movies were analyzed with a custom Fiji macro and custom Python scripts (*Verasztó, 2017*; copy archived at https://github.com/elifesciences-publications/Veraszto_et_al_2017). Correlation analyses were created using Fiji and a custom Python script (*Verasztó et al., 2017*).

## Acknowledgements

The research leading to these results received funding from the European Research Council under the European Union's Seventh Framework Programme (FP7/2007-2013)/European Research Council Grant Agreement 260821. The research was supported by a grant from the DFG - Deutsche Forschungsgemeinschaft (Reference no. JE 777/1). We thank Hernando Vergara for providing DAPI scans for the nuclear reference template for the 2-day-old larva and for generating the nuclear reference template for the 3-day-old larva.

## Additional information

### Funding

| Funder | Grant reference number | Author |
|---|---|---|
| Max-Planck-Gesellschaft | | Elizabeth A Williams<br>Csaba Verasztó<br>Sanja Jasek<br>Markus Conzelmann<br>Philipp Bauknecht |
| Deutsche Forschungsgemeinschaft | JE 777/1-1 | Elizabeth A Williams |

The funders had no role in study design, data collection and interpretation, or the decision to submit the work for publication.

### Author contributions

Elizabeth A Williams, Formal analysis, Investigation, Methodology, Writing—original draft, Writing—review and editing; Csaba Verasztó, Formal analysis, Investigation, Visualization, Methodology, Writing—original draft, Writing—review and editing; Sanja Jasek, Markus Conzelmann, Réza Shahidi, Resources, Data curation, Investigation, Visualization, Methodology, Writing—review and editing; Philipp Bauknecht, Resources, Data curation, Investigation, Methodology, Writing—review and editing; Olivier Mirabeau, Conceptualization, Data curation, Software, Formal analysis, Investigation, Visualization, Methodology, Writing—review and editing; Gáspár Jékely, Conceptualization, Data curation, Formal analysis, Supervision, Funding acquisition, Visualization, Methodology, Writing—original draft, Project administration, Writing—review and editing

### Author ORCIDs

Elizabeth A Williams (iD) https://orcid.org/0000-0003-3067-3137
Csaba Verasztó (iD) http://orcid.org/0000-0001-6295-7148
Gáspár Jékely (iD) http://orcid.org/0000-0001-8496-9836

### Decision letter and Author response

Decision letter https://doi.org/10.7554/eLife.26349.041
Author response https://doi.org/10.7554/eLife.26349.042

## Additional files

### Supplementary files

• Supplementary file 1 .xls file containing ANS synaptic connectivity spreadsheet, node ID to cell ID key, chemical network parameters, and log10 normalized expression values from mapping of single cell data for neuropeptides, GPCRs, sensory genes and neurotransmitter synthesis enzymes in separate worksheets.
DOI: https://doi.org/10.7554/eLife.26349.036
• Transparent reporting form
DOI: https://doi.org/10.7554/eLife.26349.037

### Major datasets

The following previously published dataset was used:

| Author(s) | Year | Dataset title | Dataset URL | Database, license, and accessibility information |
|---|---|---|---|---|
| Achim K, Pettit JB, Saraiva LR, Gavriouchkina D, Larsson T, Arendt D, Marioni JC | 2015 | High-throughput spatial mapping of single-cell RNA-seq data to tissue of origin. | https://www.ebi.ac.uk/arrayexpress/experiments/E-MTAB-2865/ | Publicly available at ArrayExpress (accession no. E-MTAB-2865) |

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
