## [Decision Letter]

Thank you for submitting your article "Synaptic and peptidergic connectome of a neurosecretory centre in the annelid brain" for consideration by *eLife*. Your article has been reviewed by two peer reviewers, and the evaluation has been overseen by Oliver Hobert as the Reviewing Editor and Eve Marder as the Senior Editor. The reviewers have opted to remain anonymous.

The reviewers have discussed the reviews with one another and the Reviewing Editor has drafted this decision to help you prepare a revised submission.

This paper reports on the peptidergic and synaptic connectome of a whole neurosecretory centre in the 3-days old larva of an annelid (*Platynereis dumeriliii*). This work is not only relevant for annelid biology but also informing vertebrate and insect neuroendocrinologists, in light of the apparent conservation of neurosecretory centres in Bilateria. The special strength of the study lies in the small size of the system, which allows systemic insight into the interplay between paracrine and synaptic networks that would be difficult or impossible to get in other models.

Williams and colleagues take advantage of a combination of existing resources and available techniques to outline the connectivity and information transfer between the peptidergic and synaptic components in the *Platynereis* forebrain. They reconstruct cellular projections and ultrastructure from an existing volume TEM dataset. They survey a large collection of identified neuropeptides and de-orphanized receptors. They use published single cell transcriptomic data to identify cells expressing these neuropeptides and/or their receptors. Finally, they determine the location of selected peptidergic neurons by in situ hybridization and correlate with cellular morphologies via concomitant antibody staining (acetylated tubulin).

While this manuscript has good potential, there are a number of issues that the authors need to address in a satisfactory manner before the manuscript can be considered for publication.

1) One overarching issue is the requirement for a better integration of the existing work with previous work on which this work is largely built. Previous work should be more clearly acknowledged and it should be more explicitly addressed what is novel and what has been known before. Specifically:

- The details of much of the connectomics are already in: doi: https://doi.org/10.1101/108035.

- The data of the transcript expression are from a different, published manuscript (Achim et al. (2015) Nature Biotechnology 33(5), pp. 503- 509.)

- The peptide localization by antibodies in relation to the connectome and deorphanization of receptors have all been already published by the Jekely lab (e.g. doi: http://dx.doi.org/10.7554/*eLife*.11147 and doi: 10.1016/j.celrep.2015.06.052.).

The authors need to be more explicit in addressing how the present work extends the previous work.

2) There are a few loose ends in terms of points initially raised by the authors but then not further pursued in the Results or Discussion: For example, in the Introduction the authors motivate their studies by the evolutionary comparison between the worm's 'apical nervous system' (ANS) and neurosecretory forebrain centers in other bilaterian animals. Have new conclusions be drawn? In either case this should be discussed.

3) One caveat for the study is the discrepancy in stage between the in situ stainings and the single-cell transcriptomic data, and the volume TEM resource, which have been established for 2 and 3 days of development, respectively. Apparently many of the differentiated morphologies stay the same; yet, this remains a challenge. It is not clear why in situs are shown for 2 days only; to efficiently bridge the 2- and 3-days dataset close-ups should best be shown for 3-days, too.

4) Relating to this, how ambiguous is the mapping of cells based on ciliary morphology. How do you align EM ciliary morphology with confocal reconstructions? Does the proliferation of the brain between 2 and 3 days affect number and position of the ciliated cells? It would be useful to compare both stages more thoroughly to substantiate the mapping, maybe by providing overviews for 2 and 3 days brains side- by-side with a tentative labeling of corresponding cells.

5) And in more detail, please provide at least one or two examples of the same neuron and cilium in a 3D confocal reconstruction and in an EM-based reconstruction.

6) The projection interneurons connecting the peptidergic system to the synaptic system (subsection “Projection interneurons connect the peptidergic ANS to the synaptic nervous system”) are still peptidergic themselves. Is there evidence for peptidergic synaptic transmission, or how do you distinguish between dense core vesicles in a peptidergic synapse and in paracrine release? Figure 2 is meant to show a peptidergic synapse. Maybe it could be complemented with an example of dense core vesicle paracrine release?

7) The authors should be very clear the *Platynereis* nervous system at these stages is actually a combination of larval and adult features (while the ciliated bands are only a larval feature). This concerns in particular several of the apical cells. So which of the cell types/ connectivity features do the authors believe are really relevant for larval behaviors and which rather for the adult worm? There is a whole developmental dimension that the authors completely ignore.

8) The authors still stick to the view that the pars intercerebralis of insects has a common ancestry with the vertebrate hypothalamus and the medial apical domain of invertebrate larvae. There is highly informative more recent work on the centipede Strigamia (Dev Biol. 2014 Dec 1;396(1):136-49. doi: 10.1016/j.ydbio.2014.09.020. Epub 2014 Sep 26.) that suggest that this is incorrect.

---

## [Author Response]

While this manuscript has good potential, there are a number of issues that the authors need to address in a satisfactory manner before the manuscript can be considered for publication.1) One overarching issue is the requirement for a better integration of the existing work with previous work on which this work is largely built. Previous work should be more clearly acknowledged and it should be more explicitly addressed what is novel and what has been known before. Specifically:- The details of much of the connectomics are already in: doi: https://doi.org/10.1101/108035.- The data of the transcript expression are from a different, published manuscript (Achim et al. (2015) Nature Biotechnology 33(5), pp. 503- 509.)- The peptide localization by antibodies in relation to the connectome and deorphanization of receptors have all been already published by the Jekely lab (e.g. doi: http://dx.doi.org/10.7554/eLife.11147 and doi: 10.1016/j.celrep.2015.06.052.).The authors need to be more explicit in addressing how the present work extends the previous work.

We cite the papers from which we obtained the TEM data, single cell transcriptome data, and de-orphanized receptor data when we refer to these datasets in the text.

Regarding the connectome data, while the scanned TEM sections were already available, the ANS region of the connectome was not previously reconstructed and all neurons in this region were traced, reviewed and reconstructed during the current work published here, thereby extending our analysis of the existing TEM dataset.

Regarding the singe cell transcriptome data, we downloaded the previously published raw RNA-Seq data, and mapped this data to our own *Platynereis* reference transcriptome, therefore all downstream data analysis was carried out specifically for this project, including the projection of gene expression patterns onto the 2D single cell spatial map. We used the previously published spatial predictions of Achim et al. to provide a basis from which to construct our 2D single cell spatial map.

In addition to the already published deorphanized receptors, we established three new deorphanized receptors through biochemical assays for this project. We used information about the previously published deorphanized receptors and the newly deorphanized receptors to establish our maps of neuropeptide signalling networks in the 2D single cell spatial map.

We have updated the sections describing these datasets in the text so that the work we have done here, and the work that was previously published, is clearer.

2) There are a few loose ends in terms of points initially raised by the authors but then not further pursued in the Results or Discussion: For example, in the Introduction the authors motivate their studies by the evolutionary comparison between the worm's 'apical nervous system' (ANS) and neurosecretory forebrain centers in other bilaterian animals. Have new conclusions be drawn? In either case this should be discussed.

We have now modified our Introduction, Results and Discussion to provide better flow regarding the concept of comparing the *Platynereis* ANS to other neurosecretory forebrain centers. Here, we have added to the list of neuropeptides known to be expressed in the *Platynereis* ANS; previously only FMRFa and vasotocin neuropeptides were known to be expressed in this region (Tessmar-Raible et al). The novel finding here is that signalling within the *Platynereis* ANS, occurs mainly through neuropeptidergic channels and not synaptic connections. This suggests that local/short-range neuropeptidergic signalling has an important role to play within anterior/forebrain neurosecretory centers, perhaps in neuronal activity modulation, rather than only being a means of long-range, hormonal peptidergic signalling between these centers and the rest of the body.

To gain an evolutionary insight into neurosecretory forebrain centers, we have now additionally carried out a comparative analysis of scRNA-seq data from cells in the mouse hypothalamus (Romanov et al. 2016, Campbell et al. 2017) to the *Platynereis* scRNA-Seq dataset from Achim et al. 2015. This was a collaborative work with Dr Olivier Mirabeau, whom we now include as an author on the paper. We could identify a co-expression signature for mouse hypothalamic neurons and *Platynereis* ANS neurons indicative of circadian clock endocrine cells. We could also identify conserved co-expression of different neuropeptide pairs indicative of conserved combinatorial signals from hypothalamic and ANS cells (Figure 3—figure supplement 6).

3) One caveat for the study is the discrepancy in stage between the in situ stainings and the single-cell transcriptomic data, and the volume TEM resource, which have been established for 2 and 3 days of development, respectively. Apparently many of the differentiated morphologies stay the same; yet, this remains a challenge. It is not clear why in situs are shown for 2 days only; to efficiently bridge the 2- and 3-days dataset close-ups should best be shown for 3-days, too.

We now show close-up scans of cells labelled by immunostaining or in situ hybridization in both 2 day-old and 3 day-old larvae in Figure 1—figure supplement 4 and Figure 1—figure supplement 5. We have also generated a 3 day-old neuropeptide gene expression atlas for 58 neuropeptide precursor genes (Figure 1 and Figure 1—figure supplement 3) which can be compared to the existing 2 day-old atlas to assess variation in neuropeptide precursor expression in the ANS region between 2 dpf and 3 dpf larval stages. We describe the changes in expression between 2-day-old and 3-day-old larval stages in the results. In addition, we provide close-up scans for neuropeptide precursor genes that label cells that we could identify in our TEM data (Figure 1—figure supplement 4 and Figure 1—figure supplement 5).

4) Relating to this, how ambiguous is the mapping of cells based on ciliary morphology. How do you align EM ciliary morphology with confocal reconstructions? Does the proliferation of the brain between 2 and 3 days affect number and position of the ciliated cells? It would be useful to compare both stages more thoroughly to substantiate the mapping, maybe by providing overviews for 2 and 3 days brains side- by-side with a tentative labeling of corresponding cells.

We now show anterior close-up scans of patterns of ciliation for 2-day-old, 3 day-old and 6 day-old larvae labelled by immunostaining with an antibody against acetylate tubulin to label cilia and axonal scaffold (Figure 3—figure supplement 5). The cells with unique ciliary morphology in the centre of the ANS do not proliferate between 2 day-old and 6 day-old stages, and corresponding cells can be identified across the 3 stages for several neurons. The position of these cells in the larval brain also remains relatively similar from 2 day-old to 6 day-old stages.

Rather than aligning EM and confocal reconstructions, we compare cilia morphology from 3D reconstructions generated from the two datasets (Figure 1—figure supplement 4 and 5 plus source data, and Video 2.

5) And in more detail, please provide at least one or two examples of the same neuron and cilium in a 3D confocal reconstruction and in an EM-based reconstruction.

We have now reconstructed the central 7 sensory neurons of the 3-day-old *Platynereis* ANS from our TEM dataset to generate a 3D video showing their ciliary morphology (Video 2); SN^IRP2^ (2), SN^MIP1^ (2), SN^WLD^ (2), SN^YF5cil^ (1). These can be compared to 3D videos of close-ups of these cells from larvae labelled by immunostaining or in situ hybridization of 3-day-old larvae to label the corresponding precursor genes or peptides (in situ hybridization: IRP2, WLD, bursicon B, immunostaining: FMRFa, YFa, MIP) in Figure 1—figure supplement 5—source data1.

6) The projection interneurons connecting the peptidergic system to the synaptic system (subsection “Projection interneurons connect the peptidergic ANS to the synaptic nervous system”) are still peptidergic themselves. Is there evidence for peptidergic synaptic transmission, or how do you distinguish between dense core vesicles in a peptidergic synapse and in paracrine release? Figure 2 is meant to show a peptidergic synapse. Maybe it could be complemented with an example of dense core vesicle paracrine release?

We have reconstructed a section (50 layers) of an RGWa-expressing projection interneuron, IN^RGW-dcr1^ from our TEM dataset and characterized the distribution of synapses and dense core vesicles in the axon of this neuron where it runs through the neurosecretory plexus (Figure 4—figure supplement 1, Video 3). Dense core vesicles occur throughout the axon, however no peptidergic synapses were identified, indicating that these neurons signal through peptidergic endocrine release, and synaptically with neurotransmitters.

7) The authors should be very clear the Platynereis nervous system at these stages is actually a combination of larval and adult features (while the ciliated bands are only a larval feature). This concerns in particular several of the apical cells. So which of the cell types/ connectivity features do the authors believe are really relevant for larval behaviors and which rather for the adult worm? There is a whole developmental dimension that the authors completely ignore.

We now include an additional paragraph in the Discussion discussing the role of the ANS in larval and adult *Platynereis*, and how the cellular complement of the ANS changes during development, based on what we learn from current data from this paper, and previous publications investigating *Platynereis* apical sensory cells and circadian rhythm. We conclude that the ANS plays an important role in the maintenance of circadian rhythm in both larval and adult *Platynereis*, although it is likely that this is only one of the ANS’s many functions as a sensory-neuroendocrine organ.

8) The authors still stick to the view that the pars intercerebralis of insects has a common ancestry with the vertebrate hypothalamus and the medial apical domain of invertebrate larvae. There is highly informative more recent work on the centipede Strigamia (Dev Biol. 2014 Dec 1;396(1):136-49. doi: 10.1016/j.ydbio.2014.09.020. Epub 2014 Sep 26.) that suggest that this is incorrect.

We now cite this more recent work on the centipede Strigamia regarding the evolution of insect neurosecretory centres. In the Discussion, we focus on the comparison between the *Platynereis* ANS and vertebrate hypothalamus.